# Regulatory T Cells Decreased during Recovery from Mild COVID-19

**DOI:** 10.3390/v14081688

**Published:** 2022-07-30

**Authors:** Purilap Seepathomnarong, Jomkwan Ongarj, Ratchanon Sophonmanee, Bunya Seeyankem, Sarunyou Chusri, Smonrapat Surasombatpattana, Nawamin Pinpathomrat

**Affiliations:** 1Department of Biomedical Sciences and Biomedical Engineering, Faculty of Medicine, Prince of Songkla University, Songkhla 90110, Thailand; 6310320007@psu.ac.th (P.S.); jkwanp.o@gmail.com (J.O.); 6010210367@psu.ac.th (R.S.); 6210320014@psu.ac.th (B.S.); 2Department of Internal Medicine, Faculty of Medicine, Prince of Songkla University, Songkhla 90110, Thailand; sarunyouchusri@hotmail.com; 3Department of Pathology, Faculty of Medicine, Prince of Songkla University, Songkhla 90110, Thailand; pornapat.s@psu.ac.th

**Keywords:** regulatory T cells, SARS-CoV-2, COVID-19, immune response, cytokine production, PMBC, viral peptides, inflammatory response

## Abstract

Depending on the intensity and duration of SARS-CoV-2 infection, the host immune response plays a significant role in immunological protection. Here, we studied the regulatory T-cell (Treg) response in relation to kinetic change and cytokine production in patients with mild COVID-19. Nineteen SARS-CoV-2-positive patients were recruited, and blood was collected at four time points, i.e., seven days after admission, after discharge, and one and three months after recovery. CD3^+^CD4^+^CD25^+^CD127^low^ was marked as the Treg population, with IL-10 and TGF-β used to study cytokine-producing Tregs. IFN-γ-producing CD8^+^ T cells were observed for an effector response. The Treg percentage in patients with mild COVID-19 increased during hospitalization compared to during the recovery period. Peripheral blood mononuclear cells (PBMCs) were quantified, and the T-cell response was characterized by re-stimulation with S1 and N peptides. IL-10 and TGF-β were produced by CD25^+^CD127^low^ T cells during the active infection phase, especially with N peptide stimulation. Compared to N peptide stimulation, S1 peptide stimulation provided superior IFN-γ-secreting CD8^+^ T-cell responses. Our results suggest that while IFN-γ^+^CD8^+^ T cells confer antiviral immunity, cytokine-producing Tregs may have a substantial role in regulating inflammatory responses in mild SARS-CoV-2 infection. Novel vaccine development may also consider enhancing T-cell repertoires.

## 1. Introduction

On March 11, 2020, the World Health Organization (WHO) declared COVID-19 to be a novel infection. The infection is caused by severe acute respiratory coronavirus 2 (SARS-CoV-2), a virus that causes respiratory diseases [1,2]. According to the WHO, SARS-CoV-2 has now become a pandemic, with over 526 million confirmed cases and more than six million deaths since December 2019 [3]. Effective vaccines have been allocated to prevent the spread of the disease; however, we still need to elucidate the host immune response to this virus [4,5]. 

Despite the large number of cases and deaths, information on the presence and phenotype of SARS-CoV-2-specific T cells is unclear [6]. The clinical deterioration of SARS-CoV-2-infected patients is perhaps due to increased systemic cytokine levels, known as a cytokine storm. Simultaneously, an adequate number of natural regulatory T cells (nTreg) and induced regulatory T cells (iTreg) are stimulated in the infected pulmonary tissue to reduce excessive inflammation and repair the tissue [7]. However, regulatory T cells (Tregs) were significantly decreased in some hospitalized COVID-19 patients, which might indicate a immunosuppression deficiency [8]. Therefore, the study of Tregs may provide a better understanding of clinical interventions to rebuild immune tolerance for viral pneumonia. 

Here, we observed the kinetic change in Treg responses as well as cytokine production by the Tregs of SARS-CoV-2-infected patients. The percentages of CD25^+^CD127^low^ (Treg) and CD25^+^CD127^low^FoxP3^+^ (FoxP3Treg) were observed during the hospital stay, which decreased in the recovery phase. The IL-10 production by Tregs increased when patients were newly infected with the virus and was comparable when the infection was resolving. Effector CD8^+^ T cells were also analyzed. Effector CD8^+^ T cells were absent at one month of recovery but were detected in the late recovery phase. This IFN-γ^+^-secreting CD8^+^ T-cell population was also observed in uninfected donors.

## 2. Materials and Methods

### 2.1. Ethical Approval

The ethical code was submitted to the Office of Human Research Ethics Committee (HREC), Faculty of Medicine, Prince of Songkla University, under the REC.63-092-4-1. 

### 2.2. Recruitment and Informed Consent 

Patients with mild COVID-19 who were infected with SARS-CoV-2 but who did not have viral pneumonia were recruited for the study. The patient was informed that he/she was diagnosed with COVID-19. This patient information disclosure method was authorized by the head of the Department of Pathology. A trained nurse in the COVID-19 ward provided the patient with an information sheet to introduce the project. The principal investigator or the sub-investigators spoke to the patient through a telephone outside the isolation room, answered queries, and received verbal consent from the patient. After the patient was clear of the infection, written consent was signed in the presence of the principal investigator or a sub-investigator. 

### 2.3. Variants

SARS-CoV-2 variants of concern were randomly screened by the Department of Pathology. The cohort was carried out over the duration of the pandemic. The alpha strains, which were the cause of the first wave of the pandemic, were B.1.113 and B.6. Beta (B.136.16) and delta strains were the cause of the second and third waves.

### 2.4. Blood Collection and Transportation

Blood collection was performed within seven days of admission to the hospital. The following collections were completed on the day of discharge as well as one month and three months after discharge. The collected blood (20 mL) was divided into three heparinized tubes and one clot blood tube and packed into a Ziploc bag to be placed into a transport container labeled “biohazard”. The blood samples were processed within 8 h of collection.

### 2.5. Separation of PBMCs

The collected blood in the heparinized tubes was pooled in a 50 mL tube and centrifuged at 836× *g* for 5 min at 22 °C with a break. The plasma (2 mL) was collected in two 1.5 mL screw cap tubes and stored at −80 °C. The remaining blood was resuspended in RPMI (Gibco) to a final volume of 30 mL and gently poured into a new 50 mL tube on top of 15 mL of Lymphoprep and centrifuged at 836× *g* for 30 min at 22 °C without a break. The floating band between Lymphoprep and RPMI was collected using a pasture pipette and washed twice in a new 50 mL tube and topped up with RPMI to 40 mL. It was then centrifuged at 836× *g* for 10 min at 22 °C with a break after the first wash and at 301× *g* for 10 min at 22 °C with a break after the second wash. The pellet was resuspended with 3 mL of RPMI, and the cell number was counted with trypan blue 1:10 (cell:dye) using counting slides under a light microscope. The cell suspension was topped with RPMI up to a final volume of 20 mL and centrifuged at 470× *g* for 5 min at 22 °C with a break. The cell pellet was then diluted to 3 × 10^6^ cells/mL with freezing media (filtered FBS + 10% DMSO). The cryovials were filled with 1–1.5 mL/vial of cell suspension and cooled in a Mister Frosty (Bio-Center) before being transferred to a liquid nitrogen tank. 

### 2.6. Surface Staining and Intracellular Staining (ICS)

The 1 × 10^6^ PBMCs were seeded into a 96-well plate and washed with R10, followed by spinning for 5 min at 470× *g* and 22 °C. The pellet was suspended in 150 μL of supplemented R10 (CD28 and CD49d in R10). Each well was separately stimulated with 50 μL 2 μg/mL N (Miltenyi) (Appendix A) and S1 (ProImmune) (Appendix A) peptides diluted in supplemented R10 for 2 h at 37 °C and 5% CO_2_. Then, 20 μL of GolgiPlug (BD) was added, and the samples were incubated at 37 °C and 5% CO_2_ for 16 h. After this, the plates were spun for 5 min at 470× *g* and 22 °C. The supernatant was collected separately at −20 °C and washed with 200 μL of PBS/BSA and spun at 470× *g* and 22°C for 5 mins. Live/dead staining with a working concentration of FVS700 at 1:1000 (1 μL in 999 uL PBS/BSA) was added to a 75 μL/well and incubated at 37 °C with 5% CO_2_ for 7 min. All of the wells with samples were washed with 200 uL of PBS/BSA and spun at 470× *g* and 22 °C for 5 min. A surface antibody cocktail (CD3- PerCP, CD4- APC-A750, CD8-APC, CD25-FITC, CD127-Violet 660) (75 μL/well) was then added to the live/dead stained cells and incubated at 4 °C for 30 min in the dark (Appendix A). After surface staining, the plate was washed with 200 μL of PBS/BSA and then spun at 470× *g* and 22 °C for 5 min. To fix and permeabilize PBMCs, Fix/Perm (BD) (100 μL/well) was added to ICS wells and incubated at 4 °C for 20 min in the dark. Before carrying out ICS, the plate was washed with 200 of μL perm/wash buffer (BD) and spun at 470× *g* and 22 °C for 5 min. To stain intracellular cytokines and the transcriptional factor inside the cells, an ICS antibody cocktail was added (ICS antibodies were diluted in perm/wash buffer, IFN-Gamma- PC7, TGF-Beta- PB450, IL-10- PE, FoxP3- ECD) (Appendix A). The plates were incubated at 25 °C for 30 min in the dark. The cells were washed twice and resuspended in 50 μL of FACS buffer for analysis on a CytoflexS Beckman. The acquired data were gated using FlowJo Software (Version 10) (Appendix A).

### 2.7. Statistical Analysis

Statistical analyses were performed using GraphPad Prism 9 software (GraphPad Software Inc., San Diego, CA, USA). To define the statistical significance, the Mann–Whitney U test was used to compare two groups, while the Kruskal–Wallis test followed by Dunn’s multiple-comparison test were used when analyzing multiple groups. Values of *p* ≤ 0.05 were considered statistically significant. * *p* ≤ 0.05.

## 3. Results

### 3.1. Study Participants

This study was conducted at the Songklanagarind Hospital, Prince of Songkhla University, in Songkhla, Thailand. The demographics of the participants are shown in Table 1.

### 3.2. Regulatory T Cells in COVID-19 Patients

Tregs are a subpopulation of CD4+ T cells indicated with CD25^+^CD127^low^ (Figure 1A,B). FoxP3 is the transcriptional factor for Tregs and has been used to specify the Treg population (Figure 1C). In our study, there was an increase in the Treg response after stimulation with the N peptide of SARS-CoV-2 at the discharge time point compared to 1 month after recovery. However, the difference was not statistically significant (Figure 1D-1). The percentage of FoxP3+ Tregs after stimulation by the N peptide of SARS-CoV-2 at the time of admission was four-fold compared to 1 month and 3 months after recovery, with *p* = 0.0145 and 0.0318, respectively (Figure 1D-2). The kinetic change in the Tregs was high at the admission and discharge time points and was the lowest at 1 month after recovery when stimulated by both N and S1 peptides. However, the response generated towards S1 peptide stimulation was higher than it was for N peptide stimulation. (Figure 1E). 

During early admission, the percentage of Tregs in the T-cell population that were stimulated by the S1 peptide was relatively increased compared to the percentage of this cell population at 1 month after recovery, but it was not statistically significant (Figure 1F-1). The percentage of FoxP3+Tregs that were stimulated by the S1 peptide was similar among the different time points (Figure 1F-2). FoxP3 regulatory T cells increased during the admission and discharge time points and reduced at 1 month after recovery on stimulation with the S1 peptide. The kinetic change in the FoxP3+Tregs stimulated with the N peptide was relatively high during the first time point compared to the later time points and gradually decreased as it progressed towards the last time point. Comparing the responses obtained from different stimulations, the SARS-CoV-2 S1 peptide stimulation consistently provided higher responses compared to the N peptide stimulation (Figure 1G). 

### 3.3. Cytokine-Producing CD25^+^CD127^low^ T Cells (Tregs)

To gain a better understanding of the T-cell responses during COVID-19, Tregs were gated to analyze the produced cytokines: TGF-β and IL-10 (Figure 2 and Figure 3).

#### 3.3.1. TGF-β-Producing CD25^+^CD127^low^ T Cells (Tregs)

TGF-β-producing Tregs produce an anti-inflammatory cytokine that aids in the tissue repair process. To analyze TGF-β-producing Tregs, T cells were gated to see CD25^+^CD127^low^ (Figure 2A), and CD25^+^CD127^low^ T cells were continuedly gated to see TGF-β+ Tregs (Figure 2B).

The TGF-β+ Tregs that were stimulated with the SARS-CoV-2 N and S1 peptides did not vary significantly among different time points (Figure 2C,D). The kinetic change in the TGF-β+ Tregs with SARS-CoV-2 N peptide stimulation was higher than that with S1 peptide stimulation (Figure 2E). In S1 peptide stimulation, the trend of the TGF-β+ Treg response was relatively low at the discharge time point but gradually increased at 1 month and 3 months after recovery. In contrast, N peptide stimulation was relatively high during the discharge phase but declined slightly during the recovery phase (Figure 2E).

#### 3.3.2. IL-10-Producing CD25^+^CD127^low^ T Cells (Tregs)

IL-10-producing Tregs produce cytokines as an immune-inhibitory mechanism to COVID-19. To study the IL-10 produced by Tregs, T cells were gated to obtain CD25^+^ CD127^low^ T cells (Figure 3A), which were then gated to observe IL-10+ Tregs (Figure 3B).

IL-10+ Tregs stimulated with SARS-CoV-2 N and S1 peptides were not very different among the varying time points (Figure 3C,D). The kinetic change in IL-10+ Tregs by SARS-CoV-2 N peptide stimulation was higher than that by S1 peptide stimulation at every time point (Figure 3E). When PBMCs were stimulated with the SARS-CoV-2 S1 peptide, the graph of the IL-10+ Tregs was the lowest at the discharge time point but increased and then slightly decreased at the 1- and 3-month recovery time points, respectively. In the SARS-CoV-2 N peptide, the graph was high at the first time point and at discharge but gradually reduced at the 1- and 3-month recovery time points (Figure 3E).

### 3.4. IFN-γ-Producing T cells

CD8+ T cells produce effector cytokines to control viral infection and enhance antiviral responses [9,10]. To analyze IFN-γ-producing CD8+ T cells, CD8+ T cells were gated to see IFN-γ in Figure 4A,B. CD4+ T cell responses were shown in Appendix A.

During the SARS-CoV-2 N peptide stimulation, there was an increase in the production of IFN-γ-producing CD8+ T cells of around six-fold at the admission and discharge time points compared at 1 month after recovery, with *p* = 0.0397 and *p* = 0.0210, respectively (Figure 4C). The same trend was observed during SARS-CoV-2 S1 peptide stimulation. The kinetic change in IFN-γ-producing CD8+ T cells was the highest at the admission time point, dramatically reduced at discharge, and the lowest at 1 month after recovery but increased at the last time point (Figure 4D). Along with the kinetic changes in the IFN-γ CD8+ T cells, the stimulation with the SARS-CoV-2 S1 peptide was higher than that of the stimulation with the N peptide (Figure 4E). Additionally, IFN-γ-secreting CD8+ T cells were also detected in the healthcare donors with N and S1 of SARS-CoV-2 stimulation (Figure 4C,D).

## 4. Discussion

The immune response in COVID-19, caused by a new coronavirus named SARS-CoV-2, still needs to be explored. There is little knowledge of the cellular immunity that is involved in the specific immune response of patients with COVID-19 [11]. A focus on basic but important information about T-cell responses to this novel virus is needed [12]. Regulatory T cells, also known as Tregs, are a subpopulation of T cells that regulates the homeostasis of immune responses that have anti-inflammatory effects and that secretes anti-inflammatory cytokines [13,14]. Both the T-cell response and cytokine levels are different in mild and severe SARS-CoV-2-infected patients as a consequence of alternative therapeutic strategies [15,16]. There may be some immunological factors that play a critical role in the severity of infection. Therefore, in this study, the kinetic change in regulatory T-cell responses and its secreted cytokines in patients with mild COVID-19 were reported.

Tregs are the most rapidly dividing cells among all of the T-cell subsets. They inhibit pathogenic activity and maintain immune balance [17]. In this study, we found that during an active infection, patients with mild COVID-19 had a higher Tregs and FoxP3+ Tregs population. Conversely, in patients admitted to the intensive care unit with severe COVID-19, Sadeghi et al. discovered that the number of circulating Tregs was reduced [14]. In addition, they reported fewer, mostly induced, Tregs in patients with severe COVID-19, implying that Tregs play important roles in disease progression and immune system control [18]. Interestingly, the patients with mild COVID-19 analyzed in our study had a higher number of Tregs during early infection. Moreover, Tregs may play a role in decreasing disease severity. Consistent with the findings of Tahmasebi et al., patients with mild COVID-19 showed a higher percentage of Tregs (3.295% ± 1.262%) compared to those with severe infection (2.525% ± 1.219%) [19]. The study of Treg subsets, including natural Treg (nTreg), induced Treg (iTreg), and follicular regulatory Treg (Tfr), are required to obtain a better understanding of Tregs and their function.

The interactions between TGF-β and viral infections are known to be complex and dependent on illness and pathogen conditions [20,21]. A low number of Tregs and their factors FoxP3, IL-10, and TGF-β and inflammation can be regarded as the major causes of disease pathogenesis in patients with COVID-19 [14]. In this study, we found that TGF-β+ Tregs stimulated by the SARS-CoV-2 N peptide peaked when the infection was clear. This result suggests that TGF-β-producing Tregs may influence the severity progression during early infection, in contrast to the presumptions of previous studies. 

IL-10-producing Tregs play a role in balancing pathogen reduction and immunopathology in viral infections [22]. In this study, IL-10+ Tregs restimulated by the SARS-CoV-2 N peptide during the time of hospitalization were observed and slightly reduced in progressive stages. Similarly, Abers et al. discovered a significant increase in IL-10-producing Tregs in the blood of patients with severe COVID-19 compared to those with the moderate and mild forms of the disease [23]. This result suggests that IL-10 may be involved in severity progression. Because of the consistency of the increased IL-10 level in patients with severe COVID-19, IL-10 may be used for immediate cytokine evaluation for clinical manifestation and disease progression [24].

In the case of influenza virus infection, a respiratory virus infection, protective cellular immunity decreases substantially within 3 months of infection. However, antigen-specific memory cells persist at high frequencies in the spleen and lymph nodes for over a year after infection [25]. This can be explained by the fact that during the CD8+ T-cell response against infection, there are three phases of the CD8+ T-cell repertoire, namely the initial activation and expansion phase, the contraction or death phase, and the establishment and maintenance of memory phase [26]. In this study, a trend of higher IFN-γ+ CD8+ T-cell responses stimulated by SARS-CoV-2 S1 and N peptides was observed at admission. This finding can be attributed to the activation and expansion of CD8+ T cells during ongoing SARS-CoV-2 infection. The death phase may begin one month after recovery. As a result, the proportion of effector CD8+ T cells reduced briefly after recovery.

T-cell responses are critical for viral clearance. Cross-reactive T cells, on the other hand, are linked to less severe illnesses. The widespread availability of cross-reactive memory T-cell responses may contribute to the milder severity of H1N1 flu [13]. Consistently, this study discovered that IFN-γ+ CD8+ T-cell levels similar to those in healthcare donors were observed among the different time points in patients with mild COVID-19 after stimulation with the SARS-CoV-2 N peptide. Bertet et al. also demonstrated that PBMCs from 17 years after SARS-CoV infection that had not been exposed to SARS-CoV-2 could respond to the SARS-CoV-2 N peptide [27].

Our study has some limitations. Even though patients with mild COVID-19 were analyzed in this study, other severities were not included. The result may imply disease progression but does not predict its severity. A relatively small number of patients were recruited for analysis due to the logistics of the sample collection and biosafety facility. A larger sample size would provide a clearer picture of the kinetic change. The N and S1 peptides were obtained from the wild-type pathogen. Extensive studies on T cells with the peptides from different variants of concern could provide interesting observations regarding conserved T-cell responses. 

## Figures and Tables

**Figure 1 viruses-14-01688-f001:**
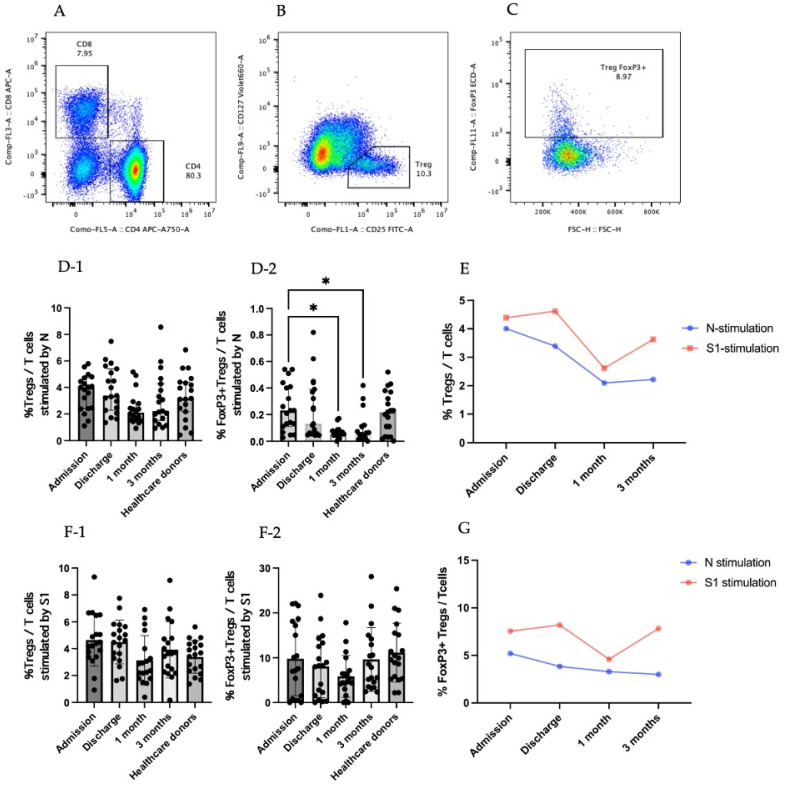
Treg responses in SARS-CoV-2-infected patients. (**A**–**C**) are gating strategies for selecting regulatory T-cell population. (**A**) CD4+ area was gated to further identify Treg cells. (**B**) CD4+ T cells were identified by CD25^+^CD127^low^ to be a Treg population. (**C**) Tregs were added by transcriptional factor FoxP3. (**D**) Results for stimulation by N peptide of SARS-CoV-2. (**D-1**) Percentage of Tregs in T-cell population. (**D-2**) Percentage of FoxP3+Tregs in T-cell population. (**E**) Percentage of Tregs in T-cell population in the form of kinetic change. (**F**) Results of stimulation by S1 peptide of SARS-CoV-2. (**F-1**) Percentage of Tregs in T-cell population. (**F-2**) Percentage of FoxP3+ Tregs in T-cell population. (**G**) Percentage of FoxP3+Tregs in T-cell population in the form of kinetic change. (**D-1**,**D-2**,**F-1**,**F-2**) The results are presented as median with 95% confidence interval (CI). Statistical significance of differences between groups was determined using the Kruskal–Wallis test followed by Dunn’s multiple-comparison test, * *p* ≤ 0.05.

**Figure 2 viruses-14-01688-f002:**
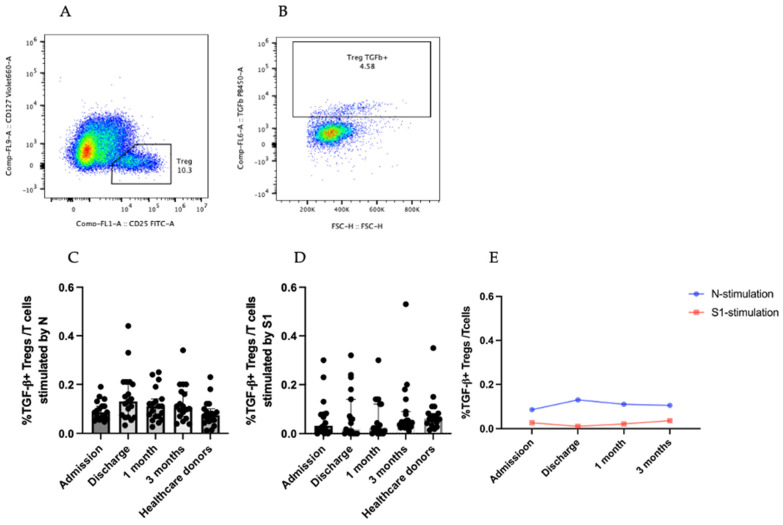
TGF-β+Treg gating strategy and population. (**A**,**B**) are gating strategies for selecting TGF-β-secreting regulatory T-cell population. (**C**) Percentage of TGF-β+ Tregs in T-cell population that were stimulated by SARS-CoV-2 N peptide. (**D**) Percentage of TGF-β+Tregs in T-cell population that were stimulated by SARS-CoV-2 S1 peptide. (**E**) Percentage TGF-β+Tregs in T-cell population in the form of kinetic change. The results are presented as median with 95% CI. Statistical significance of differences between groups was determined using the Kruskal–Wallis test followed by Dunn’s multiple-comparison test. No significant differences were observed.

**Figure 3 viruses-14-01688-f003:**
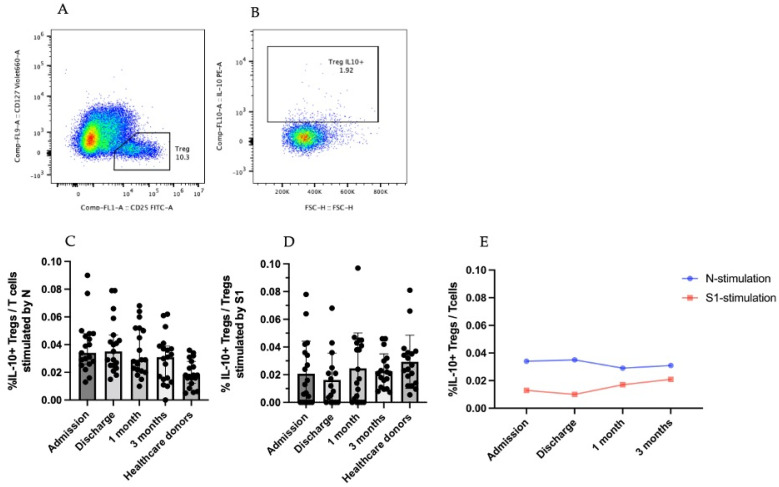
IL-10+Treg responses in SARS-CoV-2-infected patients. (**A**,**B**) are gating strategies for selecting IL-10-secreting regulatory T-cell populations. (**C**) Percentage of IL-10+Treg in T-cell population stimulated by SARS-CoV-2 N peptide. (**D**) Percentage of IL-10+Treg in T-cell population stimulated by SARS-CoV-2 S1 peptide. (**E**) Percentage IL-10+Tregs in T-cell population in the form of kinetic change. The results are presented as median with 95% CI. Statistical significance of differences between groups was determined using the Kruskal–Wallis test followed by Dunn’s multiple-comparison test. No significant differences were observed.

**Figure 4 viruses-14-01688-f004:**
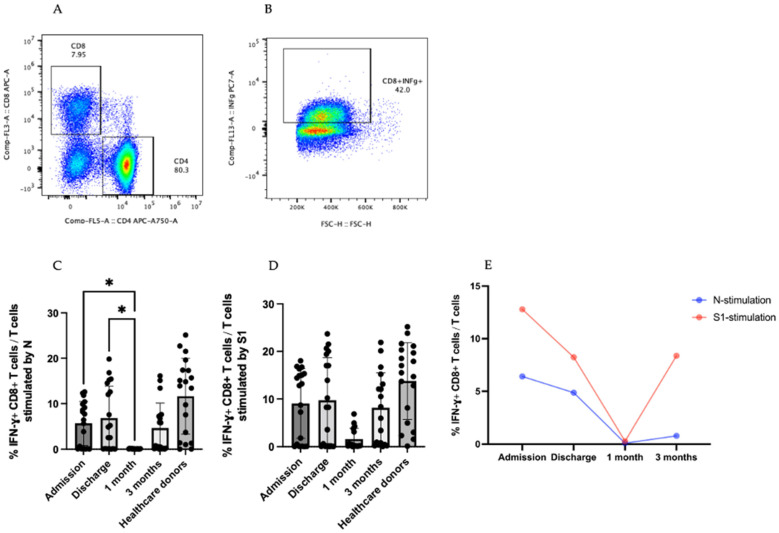
IFN-γ + T-cell responses in COVID-19 patients. (**A**,**B**) are the steps of the gating strategy of IFNγ+CD8+ T cells (**B**). (**C**,**D**) are the percentage of IFN-γ +CD8+ T cells in T-cell population stimulated by SARS-CoV-2 N and S1 peptides. (**E**) Percentage of IFN-γ+ CD8+ T cells in T-cell population in the form of kinetic change. The results are presented as median with 95% CI. Statistical significance of differences between groups was determined using the Kruskal–Wallis test followed by Dunn’s multiple-comparison test, * *p* ≤ 0.05.

**Table 1 viruses-14-01688-t001:** Baseline characteristics of mild COVID-19 patients and healthcare donors. Mild COVID-19 patients (*n* = 19) who were recruited and consented to this research were matched by sex and age with healthcare donors (*n* = 19). All patients were confirmed by positive RT-PCR results. No co-infection, immunocompromised status, immunosuppressant treatment, or endotracheal tube intubation were shown in any participants, as per the exclusion criteria. The controls were healthcare donors who had not been vaccinated or infected with SARS-CoV-2. All of the controls were seronegative for the anti-RBD IgG of SARS-CoV-2.

Baseline Characteristics	Total	Mild COVID-19 Patients	Healthcare Donors
	*n* = 38 (%)	*n* = 19 (%)	*n* = 19 (%)
Gender			
Female	18 (47.4)	9 (47.4)	9 (47.4)
Male	20 (52.6)	10 (52.6)	10 (52.6)
Mean age, years (SD)	39.4	38.8 (12.4)	39.9 (12.3)
RT-PCR positive		19 (100%)	-
Anti-RBD IgG negative		-	19 (100%)
Co-morbidity			
Co-infection		0 (0%)	-
Immunocompromise		0 (0%)	-
Receiving any immunosuppressant drug		0 (0%)	-
Endotracheal tube intubation		0 (0%)	-

## Data Availability

The data supporting the findings of this study are available from the corresponding author upon reasonable request.

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
