# Peer review of "Regulatory T Cells Decreased during Recovery from Mild COVID-19"

_viruses, 2022, doi:10.3390/v14081688_

Round 1

Reviewer 1 Report

In this paper Seepathomnarong and colleagues track the Treg proportion and phenotype of COVID-19 infected individuals across multiple time points after testing positive to SARS-CoV-2. The authors also tracked antigen-specific responses to SARS-CoV-2 S and N peptides across the multiple time points. The study cohort were only mild COVID-19 individuals with healthy controls. Overall, this brief study is informative however there are some major areas lacking in the study that warrant attention.

Line 24-26. The paper does not suggest Tregs play a role in regulating COVID-19 disease responses that reduce the severity of COVID-19 as the experiments only include mild COVID-19 donors and not severe or asymptomatic. Thus, the statement, ‘substantial role in regulating inflammatory responses that reduce COVID-19 infection severity’ should be removed. Seepathomnarong and colleagues have tracked mild COVID-19 patients and observed their Treg phenotype and function but have not linked this to COVID-19 disease progression or severity.

There was mention of nTregs and iTregs in the introduction but no further delineation of Treg subsets in the rest of the manuscript. Also, it was found by Meckiff, Cell, 2020 that follicular regulatory T cells are a predominant Treg subset in SARS-CoV-2 infection. Mention of Tfr cells and discussion of the contribution of nTregs, Tfr and iTregs in COVID-19 would be beneficial.

Line 42 suggests a large amount of Tregs are found in the pulmonary tissue of COVID-19 patients. Is there a reference to back this up?

The SARS-CoV-2 positive donors are lacking their characteristics. Only mention is they did not have ‘viral pneumonia’. A section about their demographics, i.e. ages, sex, co-morbidities is required. A clearer definition of your criteria for determining these donors had ‘mild’ COVID-19 would assist.

Why were the mild COVID-19 donors hospitalised? Usually mild disease does not require hospitalization. An explanation of why would assist the reader.

The time-point at ‘discharge’ is vague. Can you please provide a range of the length of hospital stay for these mild COVID-19 donors?

Were the COVID-19 donors vaccinated against SARS-CoV-2? If so, an acknowledgement of this as a confounding factor in the results using S peptides is necessary. Perhaps the difference in Tregs responding to S peptides are due to vaccination induced memory Tregs specific for spike epitopes.

T cell cross-reactivity between SARS-CoV-2 N and S epitopes and other human coronaviruses has been established and may account for the high responses of healthcare donors in Fig. 1 D and F. Furthermore, prior SARS-CoV-2 vaccination may alter results. Discussion of this point is helpful in interpreting the results.

Healthcare donors used as healthy controls in the experiments are not described. How many were recruited? What are their demographics?

Were the healthcare workers screened for SARS-CoV-2 infection or was there any method for assessing they are/were not infected with SARS-CoV-2?

Were the healthcare workers vaccinated against SARS-CoV-2? Since the SARS-CoV-2 vaccines are based on spike, the data using spike peptide pools should be interpreted with this in mind.

FOXP3+ Tregs were only 8.97% of CD4+ CD25+ CD127lo, Fig 1C. This is unexpectedly low. Could you provide a discussion about why?

There is no gating strategy for the intracellular markers FOXP3, IL-10, TGF-beta. Including this in the supplementary data is recommended.  

Fig 4 C-E. Is there a discussion explanation about why the IFN-gamma+ CD8s crashed to zero at one month then bounced back?

Author Response

Response to reviewer comments
Viruses
“Regulatory T cells decreased during recovery from mild COVID-19 infection”
Reviewer 1
Comments and Suggestions for Authors
In this paper Seepathomnarong and colleagues track the Treg proportion and phenotype of COVID-19 infected individuals across multiple time points after testing positive to SARS-CoV-2. The authors also tracked antigen-specific responses to SARS-CoV-2 S and N peptides across the multiple time points. The study cohort were only mild COVID-19 individuals with healthy controls. Overall, this brief study is informative however there are some major areas lacking in the study that warrant attention.

Point 1: Line 24-26. The paper does not suggest Tregs play a role in regulating COVID-19 disease responses that reduce the severity of COVID-19 as the experiments only include mild COVID-19 donors and not severe or asymptomatic. Thus, the statement, ‘substantial role in regulating inflammatory responses that reduce COVID-19 infection severity’ should be removed. Seepathomnarong and colleagues have tracked mild COVID-19 patients and observed their Treg phenotype and function but have not linked this to COVID-19 disease progression or severity.
Response1: We agreed with the reviewer. We have then amended the sentence, not to make a conclusion on severity as suggested. We have also declared this limitation on the lack of the severity groups in the discussion. 

Point 2: There was mention of nTregs and iTregs in the introduction but no further delineation of Treg subsets in the rest of the manuscript. Also, it was found by Meckiff, Cell, 2020 that follicular regulatory T cells are a predominant Treg subset in SARS-CoV-2 infection. Mention of Tfr cells and discussion of the contribution of nTregs, Tfr and iTregs in COVID-19 would be beneficial.
Response 2: We agreed with the reviewer. Our research aim is not indeed focused on the Treg subset. Therefore, we have revised the introduction as suggested. From the Xiaohong Lin et al. both nTreg and iTreg have similar phenotypic characteristics and comparable suppressive function against T cell-mediated immune responses. Thus, in this research, we focus on the phenotype of Tregs which were identified using Treg surface markers, CD25+CD127low, and Treg transcriptional factor, FoxP3+. We have mentioned the Treg subsets in the discussion as suggested.

Point3: Line 42 suggests a large amount of Tregs are found in the pulmonary tissue of COVID-19 patients. Is there a reference to back this up?
Response 3: We thank the reviewer for valuable comment. Yu Liu et al reported the majority of Tregs were found in the pulmonary tissue of COVID-19 patients. We have added the reference as advised.

Point 4: The SARS-CoV-2 positive donors are lacking their characteristics. Only mention is they did not have ‘viral pneumonia’. A section about their demographics, i.e. ages, sex, co-morbidities is required. A clearer definition of your criteria for determining these donors had ‘mild’ COVID-19 would assist.
Response 4: We thank the reviewer for helpful suggestion. We provide the patients demographic in the result Table 1 

Point 5: Why were the mild COVID-19 donors hospitalised? Usually, mild disease does not require hospitalization. An explanation of why would assist the reader.
Response 5: We thank reviewer for considerate observation. COVID-19 patients in this research infected with SARS-CoV-19 in the period of hospitalisation policy of Thailand (April 2020 - June 2021). We have added this explanation in the Recruitment and informed consent section.

Point 6: The time-point at ‘discharge’ is vague. Can you please provide a range of the length of hospital stay for these mild COVID-19 donors?
Response 6: We thank reviewer for considerate observation. Time-point at ‘discharge’ is defined as the date of the infection was cleared confirmed by RT-PCR (RT-PCR negative). 

Point 7: Were the COVID-19 donors vaccinated against SARS-CoV-2? If so, an acknowledgement of this as a confounding factor in the results using S peptides is necessary. Perhaps the difference in Tregs responding to S peptides are due to vaccination induced memory Tregs specific for spike epitopes.
Response 7: We thank reviewer for helpful observation. All COVID-19 patients in this research were not vaccinated before the infection. We confirmed with the medical record of the hospital data base. Furthermore, at that time, vaccines were shortage in Thailand and prioritised for healthcare workers. 

Point 8: T cell cross-reactivity between SARS-CoV-2 N and S epitopes and other human coronaviruses has been established and may account for the high responses of healthcare donors in Fig. 1 D and F. Furthermore, prior SARS-CoV-2 vaccination may alter results. Discussion of this point is helpful in interpreting the results.
Response 8: We thank reviewer for helpful observation. Although our healthcare donors did not vaccinate at that time. We also concern about cross-reactivity between SARS-CoV-2 N and S epitopes and other human coronaviruses. We mentioned the cross-reactive memory T-cell was reported in milder severity of  H1N1 flu and can be occurred in this research as  we have discussed the notion as suggested in line 391-398.

Point 9: Healthcare donors used as healthy controls in the experiments are not described. How many were recruited? What are their demographics?
Response 9: We thank the reviewer for helpful suggestion. We matched sex and age of the healthcare donors with the patients and provide the healthcare donors demographic in the Table1. 

Point 10: Were the healthcare workers screened for SARS-CoV-2 infection or was there any method for assessing they are/were not infected with SARS-CoV-2?
Response 10: We thank reviewer for helpful observation. All healthcare donors were unvaccinated and had no SARS-CoV-2 infection background from the medical record of the hospital database. Furthermore we also assessed anti-human IgG SARS-CoV-2 of the controls and they were all seronegative. 

Point 11: Were the healthcare workers vaccinated against SARS-CoV-2? Since the SARS-CoV-2 vaccines are based on spike, the data using spike peptide pools should be interpreted with this in mind.
Response 11: We thank reviewer for helpful observation. All healthcare donors were vaccine naïve and all seronegative for anti-human IgG SARS-CoV-2. 

Point 12: FOXP3+ Tregs were only 8.97% of CD4+ CD25+ CD127lo, Fig 1C. This is unexpectedly low. Could you provide a discussion about why?
Response 12: We thank reviewer for helpful observation. Generally, the surface phenotype of CD4+CD25+CD127low is used to quantify Treg in human PBMCs. Furthermore, FoxP3 is now accepted as a specific transcriptional factor of Tregs. We did identify FoxP3+Treg which was expected to provide a smaller number of Tregs as the staining of transcriptional factor intracellularly is difficult to achieve. More importantly, we found both of gating strategies, CD4+CD25+CD127low and FoxP3+ showed the same patterns of Treg responses. 

Point 13: There is no gating strategy for the intracellular markers FOXP3, IL-10, TGF-beta. Including this in the supplementary data is recommended.  
 Response 13: We thank the reviewer for helpful suggestion. We added the details as your recommendation in the supplementary data as suggested.

Point 14: Fig 4 C-E. Is there a discussion explanation about why the IFN-gamma+ CD8s crashed to zero at one month then bounced back? 
 Response 14: We thank reviewer for helpful observation. We believe that the reason that IFNg+CD8+ were diminished at one month because it is the death phase of CD8+ T-cell repertoires then came back because it is the establishment and maintenance of memory of CD8+ T-cell repertoires as discussed in line 383-386)

Reviewer 2 Report

In this manuscript, Seepathomnarong et al examined the different Treg responses during recovery from COVID-19 infection. They compared different cytokines producing Tregs in different recovery time point and concluded that Treg have substantial role for COVID-19 recovery. The data is interesting because the readers can learn the novel Treg responses in the COVID-19 recovery period.

The following are my specific comments.

1. In the “Material and Method”, can the author show the S1 and N peptide sequence?

2. In the Result, the author described “significant increase”, but it might be better for readers to show the fold-numbers.

3. Regarding Figure 4C, D, can the author analyze the IFN+ in SARS-CoV-2 specific T cells?

4. Do the authors know with which variants the COVID-19 exposed participants were infected? Or can the author estimate the variants based on the sample-collected date? That information would be helpful.

5. Can the authors present demographic information for all participants? If they can see the difference between age/sex, it would be better.

6. Did the authors conduct any priori sample size calculations in an attempt to gauge associated study power?

7. There are some typos.

a. Supplementary table 1. INF should be IFN

b. line39. SARS CoV-2 should be SARS-CoV-2.

c. Line45. Please delete “upper 7”

Author Response

Response to reviewer comments
Viruses
“Regulatory T cells decreased during recovery from mild COVID-19 infection”
Reviewer 2
Comments and Suggestions for Authors
In this manuscript, Seepathomnarong et al examined the different Treg responses during recovery from COVID-19 infection. They compared different cytokines producing Tregs in different recovery time point and concluded that Treg have substantial role for COVID-19 recovery. The data is interesting because the readers can learn the novel Treg responses in the COVID-19 recovery period.

The following are my specific comments.
Point 1. In the “Material and Method”, can the author show the S1 and N peptide sequence?
Response 1: We thank the reviewer for helpful suggestion. We added the details as your recommendation in the supplementary data.

Point 2. In the Result, the author described “significant increase”, but it might be better for readers to show the fold-numbers.
Response2: We thank the reviewer for helpful suggestion. We agreed the with you and change the details as your recommendation.

Point 3. Regarding Figure 4C, D, can the author analyze the IFN+ in SARS-CoV-2 specific T cells?
Response: We thank the reviewer for the suggestion. The Figure 4C, is S1 specific IFNg+ CD8+ T cells and the Figure 4D, is N specific IFNg+ CD8+ T cells. The antigen specific IFNg+ producing CD4+ T cell were analysed but not included in the manuscript. We have now included this set of data in the supplementary.

Point 4. Do the authors know with which variants the COVID-19 exposed participants were infected? Or can the author estimate the variants based on the sample-collected date? That information would be helpful.
 Response 4: We thank reviewer for helpful observation. Our study is the cohort study, so we recruited the participant in the long period. We did not examine the strains of SARS-CoV-2 in all participants. However, our hospital has been randomly screening SARS-CoV-2 variants of concern. It could be implied that the alpha strain was in the first wave of the pandemic in Thailand which were B.1.113 and B.6. Beta (B.136.16) and delta strains were the cause of the 2nd and 3rd wave of the pandemic in Thailand.

Point 5. Can the authors present demographic information for all participants? If they can see the difference between age/sex, it would be better.
Response 5. We thank the reviewer for helpful suggestion. We matched sex and age of the healthcare donors with the patients and provide the healthcare donors demographic in the Table1 

Point 6. Did the authors conduct any priori sample size calculations in an attempt to gauge associated study power?
Response 6. We thank reviewer for helpful observation. We did not perform sample size calculation. Our study is a pilot study, recruiting the participant had taken a long time due to limited cases admitted to our university hospital. We aimed to get 20 participants to obtain the preliminary information about Tregs and then could expand the study into a bigger number of patients.

Point 7. There are some typos.
a. Supplementary table 1. INF should be IFN
b. line39. SARS CoV-2 should be SARS-CoV-2.
c. Line45. Please delete “upper 7”
Response 6. We thank reviewer for helpful observation. We have revised the typos as suggested. 

Round 2

Reviewer 1 Report

Thank you to the authors for the meticulous amendments to the manuscript and the thoughtful responses to the reviewer's comments. The manuscript looks much better and I can accept it for publication. 

Author Response

We thank the reviewer for the very helpful suggestions. We are very appreciated your contribution in reviewing this manuscript.

Reviewer 2 Report

In this manuscript, Seepathomnarong et al examined the different Treg responses during recovery from COVID-19 infection. The authors answered questions and improved the manuscript. The manuscript looks better, but I still have few minor comments.

1: Regarding the Response 4, please add this information (estimated variants) in the ”Material and Method”.

2: Line3, 11, 14, 25: “COVID-19 infection” should be “SARS-CoV-2 infection”. Please fix them.

Author Response

Reviewer 2

Comments and Suggestions for Authors

In this manuscript, Seepathomnarong et al examined the different Treg responses during recovery from COVID-19 infection. The authors answered questions and improved the manuscript. The manuscript looks better, but I still have few minor comments.

Response: We thank the reviewer for your contribution in reviewing this manuscript and giving very helpful suggestions.

Point 1: Regarding the Response 4, please add this information (estimated variants) in the ”Material and Method”.

Response 1: We thank the reviewer for the suggestion. We have added the Variants in the ”Material and Method”, section 2.3.

Point 2: Line3, 11, 14, 25: “COVID-19 infection” should be “SARS-CoV-2 infection”. Please fix them.

Response 2: We thank the reviewer for the suggestion. We have changed “COVID-19 infection” to “COVID-19” or “SARS-CoV-2 infection” as suggested.